# Needs of informal caregivers of people with a rare disease: a rapid review of the literature

Julie Mcmullan ® , Lynne Lohfeld, Amy Jayne McKnight ®

Centre for Public Health, Queen's University Belfast, Belfast, UK

**Correspondence to**
Dr Julie Mcmullan;
julie.mcmullan@qub.ac.uk

## ABSTRACT

**Objectives** Many people living with a rare disease (RD) are cared for by a family member. Due to a frequent lack of individual RD knowledge from healthcare professionals, the patient and their informal caregiver are frequently obliged to become 'experts' in their specific condition. This puts a huge strain on family life and results in caregivers juggling multiple roles in addition to unique caring roles including as advocate, case manager and medical navigator. We conducted a rapid review of literature reporting on the unmet needs of informal caregivers for people living with an RD. All searches were conducted on 14 September 2021, followed by a manual searches of reference lists on 21 September 2021.

**Setting** Searches were conducted in Medline, Embase, Web of Science, GreyLit and OpenGrey.

**Results** Thirty-five papers were included in the final review and data extracted. This rapid review presents several unmet needs identified by informal caregivers of persons with an RD. The related literature was organised thematically: caregiver burden, support through the diagnosis process, social needs, financial needs, psychological needs, information and communication needs and acknowledgement from healthcare professionals.

**Conclusions** This review provides evidence that increased meaningful support is required for caregivers. Active engagement should be encouraged from this cohort in future research and awareness raised of the support available to improve the quality of life for families living with an RD. The unmet needs identified through this review will benefit people living with an RD, caregivers, healthcare professionals and policy makers.

The authors of this paper consider the unmet needs of RD caregivers a priority and recognises the value they bring to the unique caring needs.

## INTRODUCTION

In Europe, rare diseases (RDs) are those which affect less than 1 in 2000 people in a specified population.[1] Although each RD occurs infrequently, collectively RDs are a major public health issue affecting more than 450 million people globally.[2][3] RDs result in a wide variety of healthcare needs stemming

## STRENGTHS AND LIMITATIONS OF THIS STUDY

⇒ This study explored a cohort of individuals who frequently go unnoticed and unreported in the literature—informal caregivers of individuals diagnosed with a rare disease (RD).
⇒ The rapid review followed the guidelines for conducting a rapid review of the literature.
⇒ The review was conducted by experienced researchers in the area of RD carer needs who have a wealth of relevant experience. In addition, manual searches of reference lists were conducted to identify any papers not found by the initial search strategy.
⇒ It is important to acknowledge the limitations of this review. Due to resource constraints, only one author (JM) initially screened the titles and abstracts from the total set of documents retrieved. This could lead to bias.
⇒ Conducting a rapid review produced results quickly, including identifying the main self-identified needs of caregivers of people living with an RD. This is particularly important given the key role that informal caregivers play, and the fact that earlier research has focused on the person with an RD and not the caregiver.

from the involvement of multiple organ systems, such as respiratory complications, the circulatory system, muscular system, digestive system and central nervous system. Many RDs are chronic, complex and associated with physical, intellectual or neurological disabilities that significantly affect patients and their families. In addition, many families living with an RD lack peer and community support services[4-7]

Although not always the case, the family often plays a pivotal role in a person's adjustment to chronic disease and is influenced by behavioural and social factors.[8] Caregivers are defined as individuals who provide care for a person in need/care recipient. On the other hand, informal caregivers are defined as family members or close others who provide care for the person in need with no financial benefit in return. In essence, 'formal

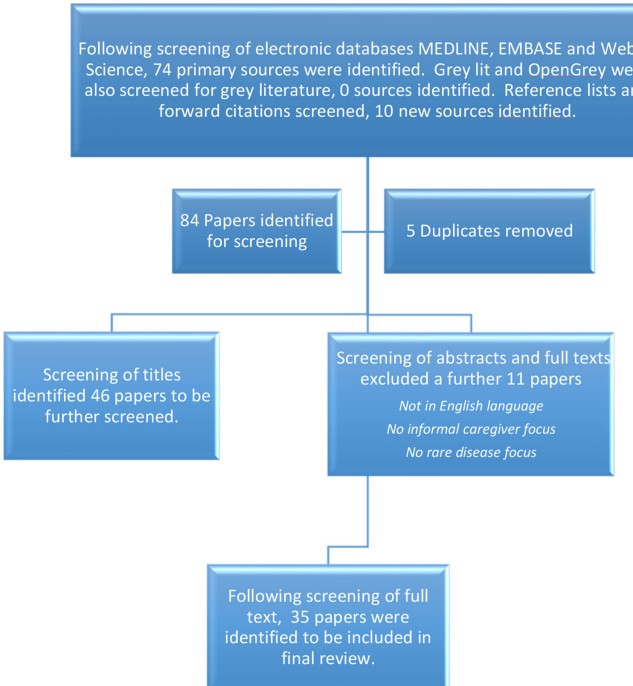

**Figure 1** PRISMA flow chart. PRISMA, Preferred Reporting Items for Systematic Reviews and Meta-Analyses.

caregivers' or 'caregivers' is used to denote a category of professionals or semiprofessionals who provide care with financial benefit. In contrast, 'informal caregivers' are typically family members or friends who provide care but have no such benefits in return (in most European countries). Caring for someone with an RD affects many areas of life including psychologically, economically, physically and logistically.[9] The importance of good mental and physical health for the informal caregiver is therefore vital for their own well-being and to ensure they can sustain the essential role of assisting the person with an RD.[10 11] Individuals with RDs and their families often have limited evidence-based information to guide decisions about disease management and symptom relief.[12 13] Further, the inherent uncertainty that comes with having an RD, including delays in diagnosis (the average time for a diagnosis of an RD in Northern Ireland is 5 years) and a lack of knowledge about current and future care needs,[14 15] impact access to services and management of the RD.[16] Research on the experience of having an RD indicates that care and service needs are often not based on the severity of the health condition. Rather, they are associated with poor quality of care and barriers to access leading to less satisfaction with healthcare services and care coordination.[15 17]

Given the few people with a specific RD, healthcare providers often are not knowledgeable about the condition. Therefore, people living with an RD and their carers have to become their own experts.[18] This causes a change to the usual patient-doctor relationship, which can bring challenges such as difficulties in communication and patients struggling to get sufficient accurate information to make informed choices.[19 20]

**Table 1** Literature search results

| Database | No of articles retrieved | No of articles included in review |
|---|---|---|
| Medline | 6 | 5 |
| Embase | 6 | 5 |
| Web of Science | 62 | 13 |
| GreyLit | 0 | 0 |
| OpenGrey | 0 | 0 |

This table shows the number of articles identified from each database before reference lists were checked.

Caring for someone with an RD can be highly demanding often requiring intense and unique care specific to the individual's needs.[21] Delayed diagnosis, lifelong caring, limited capacity for independent living, lack of treatment options and large health service needs have severe impacts on parent's physical and psychosocial well-being.[22]

Rapid reviews are an emerging type of knowledge synthesis used to inform health-related policy decisions and discussions, especially when information needs are immediate.[23–26] Rapid reviews streamline systematic review methods—for example, by focusing the literature search[23] while still aiming to produce valid conclusions. The requirements of the review, which was undertaken with a short deadline, were for a short but in-depth synthesis of the current state of the issues facing caregivers for those with an RD.

This review focuses on informal caregivers for people living with an RD. We were particularly interested in their holistic needs when caring for this unique population and how these may differ from other caring populations where there is often more support available for example, carers of people with dementia or cancer.

## METHODOLOGY
### Search strategy and inclusion criteria
Three electronic databases were searched—Medline, Embase and Web of Science—using the combined terms 'informal caregiver*' and 'rare disease*' (online supplemental file 1). All searches were conducted on 14 September 2021, with no date restrictions. Reference lists of included papers were screened for further sources. A search was also conducted of grey literature using the databases GreyLit and OpenGrey. Duplicates and non-English language articles were excluded. The criteria for inclusion were articles that address caregiving for people living with an RD. Articles on caregiving alone or RD alone were excluded. A simplified list of databases and the number of papers retrieved is shown in table 1.

### Study selection and data extraction
Database searches were last conducted on 14 September 2021 by JM. Titles and abstracts of the identified articles were downloaded onto Endnote. Duplicate articles

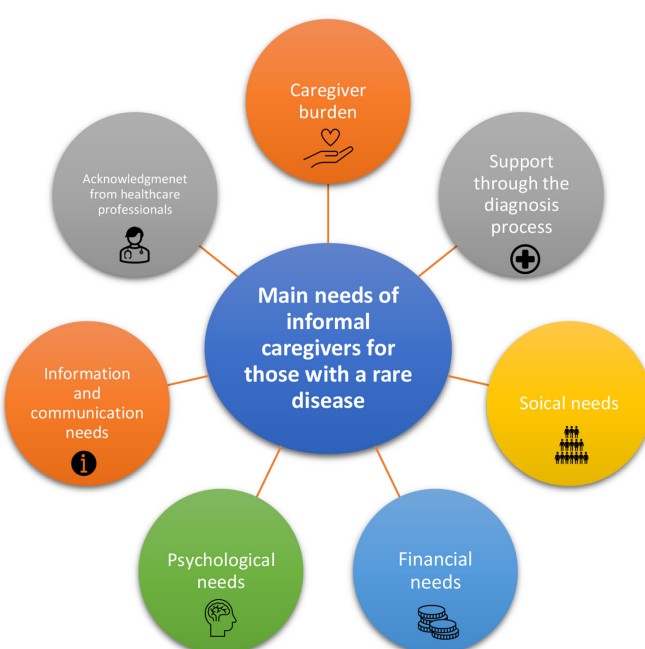

**Figure 2** Main needs of caregivers for those with a rare disease

were removed and the remaining papers were screened by analysing their titles and abstracts. In addition, manual searches of reference lists were conducted on 21 September 2021 to identify any papers not found by the initial search strategy. If relevant, the papers were then further screened by reading the full text. Data were extracted by JM who recorded the following data for each study:

► Author and year of publication.
► Country in which the study was conducted.
► Data collection method(s).
► Participants.
► Identified caregiver needs.

The identified caregiver needs were extracted from the included papers (JM) and presented in the data extraction table (AM, JM and LL) (online supplemental file 2). From this, common needs were grouped together under headings which became the themes for this review. The themes and content for the themes was agreed between the research team (AM, JM and LL).

An illustration of the search strategy including databases searched and screening methods is displayed in figure 1, modelled on the Preferred Reporting Items for Systematic Reviews and Meta-Analyses flow diagram.

### Patient and public involvement

The need for this review was highlighted by a priorities workshop in 2020, which considered contributions from >2000 individuals living with a RD (Prioritisation workshop | Rare Disease Focus: supporting rare disease throughout all communities (qub.ac.uk)). The review was also shared with representatives from a local RD charity in Northern Ireland - the Northern Ireland Rare Disease Partnership.

## RESULTS
### General description of the literature

Sources initially identified from each database were as follows: MEDLINE n=6, Embase n=6, Web of Science n=62, OpenGrey n=0 and GreyLit n=0. The screening process resulted in 30 documents which were reviewed based on their titles and abstracts. Following the screening 5 duplicate papers were removed and 25 papers were identified for full-text screening. Further studies were identified by searching the reference lists. Finally, 35 texts were included in the full review, with characteristics of each source summarised from the completed data extraction table (online supplemental file 2).

Of the texts included, publication dates were from 2006 to 2021 and the studies were conducted in Australia, Canada, Germany, Ireland, Italy, Netherlands, Spain, UK and the USA. The identified studies focus on many RDs including hereditary angioedema, Huntington's Disease, Duchenne muscular dystrophy and Von Hippel-Lindau disease.

A narrative review is presented below, highlighting the main needs of caregivers of people living with an RD. The related literature has been organised thematically under the following headings:

► Caregiver burden.
► Support through the diagnosis process.
► Social needs.
► Financial needs.
► Psychological needs.
► Information and communication needs.
► Acknowledgement from healthcare professionals.

Findings from these studies were organised thematically under the following headings as shown visually in figure 2.

### Caregiver burden

Caring for someone with an RD presents a range of challenges[27 28] that are affected by the severity of the illness and its duration, knowledge about the condition and its changes over time, and one's ability to address the emotional toll involved with long-term care.[29] Despite the stress and difficulties associated with being a family caregiver,[30] often they acknowledge the positive aspects of caring for their loved ones.[30]

The quality of life of caregivers is often compromised by their placing others' needs before their own.[31] Previous research has shown that informal caregivers do not have enough time for themselves,[32] and that caring for someone with an RD can negatively impact on all dimensions of family life.[33] Informal caregivers are often forced to miss work or school days due to the demands of their role, frequently address unpleasant events and watch a loved one suffer.[28] Consequently, they would benefit greatly from support to reduce the burden of caring for example from support groups, respite care and employment arrangements.[34] This includes both supports for informal carers of people with a variety of health issues, as well as for RD-related issues.[34]

## Support through the diagnosis process

In spite of marked medical progress over the past several decades in identifying an RD, there are substantial delays in diagnosing these conditions.[35] This can have adverse impacts on families, for example, financial and mental health.[12] Little attention is given to parent concern in the diagnostic process with previous research reporting concerns regarding how the diagnosis was delivered to parents, lack of guidance and poor follow-up post diagnosis.[35] Previous research has shown that families are often dissatisfied by the way in which a diagnosis is given, including an insensitive style of communication, not offering support or counselling, and inadequate provision of information about the disease. This can cause emotional stress for caregivers who have not yet learnt about the condition and how it will affect them and their families. Therefore, interventions to support the following a diagnosis would be useful, delivered either in hardcopy or online.[36]

## Social needs

Social isolation is common among informal caregivers with the impact on personal relationships being a reoccurring theme.[4 5] All too often, informal caregivers have little time to themselves and lack appropriate support and time for respite. This in turn causes further emotional stress and in the case of parents, uncertainty about their child's future.[37] Parents of children with an RD often feel isolated from mainstream society and struggle to stay socially connected.[4] It was suggested that the insights gained through research studies regarding the impact on caregivers' social lives should be considered in future clinical service planning. In addition, holistic, empathic, and person-centred medical and psychosocial care are urgently needed for this cohort.[6] Support is needed as relationships among family members are often impacted due to demands of caring.[4] There is a need for improved parental supportive care as many common unmet needs exist across RDs.[38]

## Financial needs

Previous research has indicated that financial issues are a top concern of caregivers of persons with RD.[6 27 30 39] The high economic costs that families must cover means that financial burden is common among caregivers of those with an RD, many of whom are forced to exhaust their financial savings.[40] Purchasing equipment, hiring professionals and the additional financial burden the illness has on families pose immense stress on family life.[41] Often financial issues impact on the ability of the caregiver to meet the individual's healthcare needs, increasing the strain on their lives.[42] Substantial social/economic burden is mostly attributable to high direct non-healthcare costs.[43] Previous research has shown that caregivers' financial burden might be conditioned by the clinical condition of the patient.[44] In addition to the direct healthcare costs, caregivers of those with an RD report that their career choices are influenced due to

their caring role. Furthermore, caring duties can result in missed working days due to emergency caring demands[37] making it difficult for many caregivers to maintain their job and advance in their careers.[41] Financial hardship adversely effects the mental health of caregivers.[42]

## Psychological needs

There is a lack of psychological support for families caring for children with an RD, and many report that accessing appropriate psychological care is difficult.[6 12] Previous research has suggested that this form of support should be offered at the time of diagnosis.[10 12] Depressive symptoms are often associated with caregiving burden and therefore there is a need to develop interventions in addition to promoting the existing validated tools for caregivers considering their special needs.[10] Caregivers often express concerns about the future,[32] experiencing emotional distress that can compromise the well-being of family carers, who attempt to maintain multiple roles.[11] Psychological interventions can help reduce stress, a sense of being burdened and feelings of isolation that many RD caregivers feel.[45] Research has suggested that screening for depression is needed and emphasises the need for a holistic approach to family mental health in the context of chronic childhood disease.[46] Peer support is a key resource in terms of information and emotional support for parents who often begin their journey feeling isolated and alone.[21]

## Information and communication needs

Suggestions were made that information about the condition should be given to caregivers at the time of diagnosis as well as signposting them to groups connecting families who share common experiences.[21] Caregivers require access to accurate information, appropriate services and improved communication between patients, families and a range of both health and social care and other public services.[2] Families require easily accessible services that include the family in the unit of care, provide support and information, and understand the process of family adjustment and adaptation in the long term.[47] Policies and structures are needed to support social and economic needs.[48] Engaging caregivers in future avenues of research is vital to ensure resources and funding are targeted in the best way.[49]

## Acknowledgement from healthcare professionals

Out of necessity, informal carers often know more about a particular RD than healthcare providers do. This often leads to poor communication and collaboration. Furthermore a lack of coordination of care force carers to fill the gap by juggling multiple roles including that of advocate, case manager and medical navigator.[50] Caregivers often experience silencing or being silenced when interacting with healthcare and social care systems and providers.[51] This has been attributed to the lack of knowledge about RDs by healthcare providers who also are unaware of the impact that caring for someone with RD.[50] This points

to the need to improve the knowledge of healthcare providers on the medical, social and financial impact these informal carers experience.[4] Healthcare providers should also acknowledge the vital role that informal caregivers play in promoting the health and quality of life of persons with an RD.[52] Caregivers, especially parents caring for someone with an RD, work hard to be heard and acquire services within health and social systems.[50] Another way that informal caregivers are silenced is due to how they are overlooked in the world of research despite growing emphasis on ensuring 'patient and public involvement'. It is vital that this pattern is changed so that they have an opportunity to be heard.[27 53]

## DISCUSSION

The purpose of this rapid review was to synthesise and describe what is currently known about the needs of informal caregivers for people living with an RD. Based on the findings, several recommendations for future healthcare practices and policies, as well as for research are evident. First, it is important to consider the extreme burden often experienced by these caregivers, many of whom place the need of others ahead of their own while navigating this journey with little or no support.[31] Support is needed to help caregivers with many aspects of their caring duties from the initial diagnosis process which has been a difficult time for many families.[35]

Social support would be welcomed to enable caregivers to have much needed time away from their caring duties to recharge and also to relieve them of the emotional strain which impacts on not only themselves but their wider family.[4] The mental strain of caregiving is evident from previous research, psychological interventions are required that consider the family as a whole to reduce the emotional strain and depression too often experienced by caregivers.[21 45] Many caregivers have also reported struggling financially not only due to the high costs of the practical side of caring but as a result of the impact caring can have on their ability to maintain a career,[41] financial support, therefore, a priority.

Clear information should be provided for caregivers from diagnosis through to their RD journey[2] to replace the current situation of confusion and uncertainty due to unclear and incomplete or conflicting messages. Lastly, caregivers must be recognised by healthcare professionals for the integral part they have in caring for someone with an RD. This relationship often changes the dynamic between health professional and patient and so caregivers must be listened to and viewed as an 'expert'.[52] All of the above-mentioned issues should be taken into consideration when planning future research, it is vital that informal caregivers views are valued.

## CONCLUSION

This rapid review presents several unmet needs identified by informal caregivers of persons with an RD. It is hoped that the findings contribute to increasing the amount of meaningful support for caregivers as well as encourage their active participation in improving the quality of life for families living with an RD. It is important that this cohort must be engaged in future research to ensure their needs are being addressed. The insights that are gained through future research concerning the impact on caregivers' social lives should be considered in the priorities and strategic directions for clinical services. Interventions to support them following a diagnosis would be useful, delivered either in hardcopy or online. Better meeting the needs of informal caregivers for people living with an RD is in the best interests of people with an RD, healthcare professionals and policy makers, as well as caregivers. It is important that awareness is raised about the range of support options that are available from health and social care providers, charities and/or support groups for informal caregivers of people with an RD.[27]

**Contributors** AJM is the guarantor of the study. JM contributed to data curation, formal analysis, data interpretation, drafting and reviewing the original manuscript. LL and AM contributed to study conception and design, data interpretation, project administration, supervision and reviewing manuscript drafts. All authors approved the final version of the manuscript for submission.

**Funding** JM was supported by an award from the NI Public Health Agency and the Medical Research Council (MC_PC_16018) – Northern Ireland Executive support of the Northern Ireland Genomic Medicine Centre though Belfast Health and Social Care Trust.

**Competing interests** None declared.

**Patient and public involvement** The need for this review was highlighted by a priorities workshop in 2020, which considered contributions from >2000 individuals living with a rare disease (RD Prioritisation workshop | Rare Disease Focus: supporting rare disease throughout all communities (qub.ac.uk)). The review was also shared with representatives from a local rare disease charity in Northern Ireland - the Northern Ireland Rare Disease Partnership.

**Patient consent for publication** Not applicable.

**Ethics approval** Not applicable.

**Provenance and peer review** Not commissioned; externally peer reviewed.

**Data availability statement** Data sharing not applicable as no datasets generated and/or analysed for this study.

**ORCID iDs**
Julie Mcmullan http://orcid.org/0000-0001-8566-4807
Amy Jayne McKnight http://orcid.org/0000-0002-7482-709X

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
