## [Reviewer comments · BMJ Open]

ARTICLE DETAILS

TITLE (PROVISIONAL)	The needs of informal caregivers of people with a rare disease: a rapid review of the literature.
AUTHORS	Mcmullan, Julie; Lohfeld, Lynne; McKnight, Amy

VERSION 1 – REVIEW

REVIEWER	Galvin, Miriam University of Dublin Trinity College, Academic Unit of Neurology, Trinity Biomedical Sciences Institute,
REVIEW RETURNED	03-Jun-2022

GENERAL COMMENTS	The needs of informal caregivers of people with a rare disease: a rapid review of the literature. This paper reports on a rapid review of literature of unmet needs of informal caregivers for people living with a rare disease. 35 papers were included in the final review. The findings evidence the need for support for caregivers and raising awareness of that need. Acknowledgement from healthcare professionals was an interesting theme. General Is it the case that the definition of rare disease as specified in the introduction is the way in which rare disease is understood in the literature you searched? How would you know the definition of rare disease used - if 'rare disease*' was a search term? Why did the authors choose a rapid review in comparison to a systematic review? Explain why getting the results quickly was important to them, why a short deadline? Supplementary file 1 has some information on conditions, it would be interesting for the reader to have some of this outlined in the text and get an indication of the rare diseases in your review results Did the authors find any differences in the needs of caregivers of different rare diseases? Even if this was not the focus of the review it could be interesting to note if some needs were linked to particular rare diseases Specific issues Introduction Page 3 Line 35 reference 11 duplicated Page 3 Line 52, could the authors detail some of the challenges as examples Page 3 Line 57 is 'parent' a typo? Methodology Was there a range on the dates included for your search? Is a search on one day sufficient to conduct a review? State why you think this is the case Manual search was mentioned in the strengths section, include that in methodology section,
---

	Results Could the authors comment on why they think there were no papers included from search of grey literature databases? Page 5, line 52 words missing? Page 6 How were the findings thematised? Discuss that process, did you use themes from the literature you found to organise them for yourself? Is this methodology or results? Social needs –parents/children mentioned a few times, was this caregiving relationship a focus or were there few findings about social needs and other kin relations? Strengths and Limitations Page 10 lines 20 'self-identified needs' did all the papers in the review include needs identified by carers themselves? Was that an inclusion criterion? Line 22-23 and the fact that earlier research has focused on the person with an RD and not the caregiver. Are there some references for this statement? Figure 2 Are these the needs that the authors themed? Or are these themes identified within the literature reviewed?
--	---

REVIEWER	Boettcher, Johannes University Medical Center Hamburg-Eppendorf
REVIEW RETURNED	19-Jun-2022

GENERAL COMMENTS	The authors present a rapid review of literature reporting on the unmet needs of informal caregivers for people living with a rare disease. The study is designed to identify existing literature on this topic and to present it in form of a narrative review. The rapid review finally included 35 papers and identified caregiver burden, support through the diagnosis process, social needs, financial needs, psychological needs, information needs, and acknowledgment from healthcare professionals as the main topics. This is an interesting paper with relevance to researchers and clinicians working in this field due to its well-structured overview of the mostly convoluted field of research. The rapid review is well presented and a good fit for the journal. I have, however, some comments to do: Minor comments and revisions  1. The research questions and aims in the introduction lack clarity and therefore should be guided by approaches like the PICO criteria. 2. In the methods section more information on the inclusion criteria should be given. 3. In the methods section the authors state that "all searches were conducted on 14 September 2021". The authors may explain in the limitations section why they did choose to not include a second search interval to include more recent research in the review. The authors also state that a manual search was conducted, but not state when this was done. 4. In the Results section an abbreviation for 'patient and public involvement' (PPI) (page 9 line 10) is given and not used again. The authors should consider removing the abbreviation. 5. In the discussion section in the sentence (page 9, line 22-25) an additional "s" is given after the word support. Please check. 6. It should be discussed whether the gender of the parents may have an impact on the presented results or if mothers and fathers tend to have different needs.
---

REVIEWER	Petrovic, Milica Catholic University of the Sacred Heart, Psychology
REVIEW RETURNED	22-Jun-2022

GENERAL COMMENTS	Dear. I had an opportunity to review your paper, and although I find it interesting and relevant to the field, I have some minor comments that I think need to be addressed before proceeding with the publication. One of my concerns is that the term "informal caregiver" is used interchangeably with the terms "family members", and/or "parents". However, informal caregivers are clearly defined in the literature and they do not necessarily mean parents or family members. Furthermore, I think it is important to define if your work is focused on informal caregivers of children living with the RD or informal caregivers of adults living with RD. If your paper is focused on the general population of informal caregivers of people living with RD regardless of the care recipient's age, then I think there is a disbalance in the introduction of the topic since on numerous occasions you address the studies and potential issues and needs of parents of children living with RD. The narrative synthesis or rather the methodology part does not clearly indicate how the final themes were obtained. I suggest making the process of synthesis more explicit by including another table where this becomes clear to the reader. Finally, you made some interesting discoveries that could be informative for policymakers and practitioners, but do not further elaborate on them nor fully include them in the implications/or strengths of your papers. I would advise you to make your contribution more explicit by adding these into the strength sections. You made some interesting suggestions regarding the possibility of creating a manual both online and hardcopy and this could be stressed out as a valuable implication for practice. I am attaching a pdf of your manuscript with some comments and suggestions that you might want to consider. Best Regards
--

VERSION 1 – AUTHOR RESPONSE

Reviewer reports:

Reviewer #1: Below you can read my comments after carefully reviewing your paper:

General

Is it the case that the definition of rare disease as specified in the introduction is the way in which rare disease is understood in the literature you searched? How would you know the definition of rare disease used - if 'rare disease*' was a search term?

Thank you for reviewing our paper and for your suggestions to improve the manuscript. We have addressed your suggestions below and hope these will make an overall improvement to the paper.

As the identified research studies were conducted in several different countries worldwide, the definition of 'rare disease' varies between studies. Although the varying attributes that lead to a diagnosis of a rare disease differ by region, the need for someone to fulfil a caring role for such individuals is the same across all definitions of such conditions. Similar challenges and issues exist

for carers no matter how a rare disease is defined and where the individual resides. Therefore, the definition of 'rare disease' in the introduction does not affect the findings of this review.

Why did the authors choose a rapid review in comparison to a systematic review? Explain why getting the results quickly was important to them, why a short deadline?

Thank you for requesting clarity around the method chosen. A rapid review was chosen for this piece of work as it was required to be completed in a timely fashion as it is associated with the development of our national rare disease action plan, leading to local policy recommendations. This review will be used as a basis of further, much needed research on this topic, prompting the development of resources to better support carers of people with a rare disease.

Supplementary file 1 has some information on conditions, it would be interesting for the reader to have some of this outlined in the text and get an indication of the rare diseases in your review results

Thank you for pointing this out, this is a welcome addition and makes the manuscript more informative. More information on the various rare diseases is now included on p6 lines 160-161. The new statement now reads,
"The identified studies focus on many rare diseases including Hereditary angioedema, Huntington's Disease, Duchenne muscular dystrophy and Von Hippel-Lindau disease."

Did the authors find any differences in the needs of caregivers of different rare diseases? Even if this was not the focus of the review it could be interesting to note if some needs were linked to particular rare diseases

This is an interesting point. There did not appear to be any differences in the needs of caregivers of different rare diseases. In fact, many similarities were found in reports of caregiver needs even when the diseases varied. This demonstrates the possibility of unmet rare disease caregiver needs across several rare diseases.

Specific issues

Introduction

Page 3 Line 35 reference 11 duplicated

Many thanks for bringing this to our attention. The duplicate reference has been removed (p3, line 73) and the new statement reads:

"can sustain the essential role of assisting the person with an RD ^{10 11}. Individuals with RDs and their,,,"

Page 3 Line 52, could the authors detail some of the challenges as examples

Thank you; this is an important issue to include. Some challenges have been added to p3 line 85-87. "...such as difficulties in communication and patients struggling to get sufficient accurate information to make informed choices ^{19 20}. "

Page 3 Line 57 is 'parent' a typo?

Thank you for noticing this, however, this is not a typo but instead refers to how these negative circumstances around care for individuals with a rare disease often impact parents of children with a rare disease who frequently undertake this caring role. We believe this is true for many carers supporting people with a rare disease, but the reference cited specifically referred to parents.

Methodology

Was there a range on the dates included for your search?

Thank you for asking this. A date range was not explicitly included in the search because the number of articles retrieved did not require additional restrictions, highlighting the lack of research in this topic.

The date range is from commencement of resources until 14th September 2021 & we have clarified that in the main text as below p4, line 109-110.

“All searches were conducted on 14 September 2021, with no date restrictions.”

**Is a search on one day sufficient to conduct a review? State why you think this is the case
Manual search was mentioned in the strengths section, include that in methodology section,**

Thank you for asking this question. The database searches were conducted on 1 day and saved. The screening process then took place, along with the manual search of references in the included articles following standard guidelines for reviews. Thanks for pointing out that this was not mentioned in the methodology section. This information has now been included on p4, line 122-124.

“Additionally, manual searches of reference lists were conducted to identify any papers not found by the initial search strategy.”

Results

Could the authors comment on why they think there were no papers included from search of grey literature databases?

Although there is much research focusing on rare disease, there is a lack of research focusing on the caring role or needs or rare disease carers. Similarly, there is a lack of relevant doctoral dissertations, government documents or policy documents focused on this area, which would explain why no papers were found and emphasises the need for this rapid review to help inform our national rare disease action plan.

Page 5, line 52 words missing?

This should have been ‘searching’, not ‘search’. This has been amended on p5, line 156 to read “...searching the reference lists. Finally, 35 texts were included in the full review, with characteristics of...”

Page 6 How were the findings thematised? Discuss that process, did you use themes from the literature you found to organise them for yourself? Is this methodology or results?

The data were extracted from each study showing the ‘identified caregiver needs’ and used to populate Table 1(Supplementary file 1: Data extraction table). The findings were then collated into common needs under umbrella terms that are displayed in figure 2.

This information belongs in the Results section which goes further to explain each theme in more detail.

Social needs –parents/children mentioned a few times, was this caregiving relationship a focus or were there few findings about social needs and other kin relations?

Most children with a rare disease are cared for by their parents. This was not an intentional finding, but it is the ‘norm’ in such situations.

Strengths and Limitations

Page 10 line 20 ‘self-identified needs’ did all the papers in the review include needs identified by carers themselves? Was that an inclusion criterion?

No; many of the studies used measures gathered from surveys or information from interviews to assess carers needs. This was not a specific inclusion criteria of the review.

Line 22-23 and the fact that earlier research has focused on the person with an RD and not the caregiver. Are there some references for this statement?

Thank you for making this point. References were added on p2, line 52 but these have now been removed given that the strengths section has to be placed prior to the introduction - (Currie & Szabo, 2019a, 2019b; Niemitz et al., 2019; Wu et al., 2020).

Figure 2 Are these the needs that the authors themed? Or are these themes identified within the literature reviewed?

These needs were themed by the authors who reached consensus on the needs based on the identified literature included in the review.

Reviewer: 2

Dr. Johannes Boettcher, University Medical Center Hamburg-Eppendorf

Comments to the Author:

The authors present a rapid review of literature reporting on the unmet needs of informal caregivers for people living with a rare disease. The study is designed to identify existing literature on this topic and to present it in form of a narrative review. The rapid review finally included 35 papers and identified caregiver burden, support through the diagnosis process, social needs, financial needs, psychological needs, information needs, and acknowledgment from healthcare professionals as the main topics. This is an interesting paper with relevance to researchers and clinicians working in this field due to its well-structured overview of the mostly convoluted field of research. The rapid review is well presented and a good fit for the journal. I have, however, some comments to do:

Minor comments and revisions

1. The research questions and aims in the introduction lack clarity and therefore should be guided by approaches like the PICO criteria.

Thank you, we appreciate this may not have been clear, so we have now structured this around the PICO criteria on p4 lines 101-105.

“This review focuses on informal caregivers for people living with a RD. We were particularly interested in their holistic needs when caring for this unique population and how these may differ from other caring populations.”

2. In the methods section more information on the inclusion criteria should be given.

Many thanks for pointing this out, there was a paragraph further down the methods section which would be better placed under the ‘inclusion criteria’ to provide the required detail. This has now been moved to p4 lines 112-115.

“The criteria for inclusion were articles that address caregiving for people living with an RD. Articles on caregiving alone or RD alone were excluded. Primary research studies and systematic review were considered for inclusion as shown in Table 1.”

3. In the methods section the authors state that “all searches were conducted on 14 September 2021”. The authors may explain in the limitations section why they did choose to not included a second search interval to include more recent research in the review. The authors also state that a manual search was conduct, but not state when this was done.

Thank you for raising this query. This rapid review was conducted to help inform our national rare disease action plan. All online searches were conducted on 14th Sept 2021, followed by manual searching of reference lists, screening, data extraction, thematic analysis and drafting the manuscript. This manuscript was submitted to BMJ Open for review on 25th March 2022 so there was no opportunity to perform a more recent search.

Again, thanks, the date has now been added to p4, line 122-124.

“Additionally, manual searches of reference lists were conducted on 21 September 2021 to identify any papers not found by the initial search strategy.”

4. In the Results section an abbreviation for ‘patient and public involvement’ (PPI) (page 9 line 10) is given and not used again. The authors should consider removing the abbreviation.

Thank you, this has now been removed.

“...ensuring ‘patient and public involvement’. It is vital that this pattern is changed so that they have an...”

5. In the discussion section in the sentence (page 9, line 22-25) an additional “s” is given after the word support. Please check.

Thank you; this was a typo and has now been removed.

“...support³¹. Support is needed to help caregivers with many aspects of their caring duties from the...”

6. It should be discussed whether the gender of the parents may have an impact on the presented results or if mothers and fathers tend to have different needs.

This is an interesting point. Most of the papers refer to ‘parents’ rather than ‘mother’ or ‘father’ so it would be difficult to say if gender impacts in this way. However from working closing with many caring charities and with rare disease collaborate groups we have discovered that the majority of informal caregivers are female (mothers, wives, sisters, etc.).

Reviewer: 3

Dr. Milica Petrovic, Catholic University of the Sacred Heart

Comments to the Author:

Dear,

I had an opportunity to review your paper, and although I find it interesting and relevant to the field, I have some minor comments that I think need to be addressed before proceeding with the publication.

- One of my concerns is that the term "informal caregiver" is used interchangeably with the terms "family members", and/or "parents". However, informal caregivers are clearly defined in the literature and they do not necessarily mean parents or family members. Furthermore, I think it is important to define if your work is focused on informal caregivers of children living with the RD or informal caregivers of adults living with RD. If your paper is focused on the general population of informal caregivers of people living with RD regardless of the care recipient's age, then I think there is a disbalance in the introduction of the topic since on numerous occasions you address the studies and potential issues and needs of parents of children living with RD.

Informal caregivers for someone with a rare disease are often family members or have a close relationship with the individual. We did not put a restriction on the age of those to be included in our review. We appreciate there may be some differences between the various age groups but found there were many more similarities and a common set of needs than differences. We therefore feel the needs we have identified are applicable to a wide span of ages. We were particularly interested in the overall or holistic needs of those caring for someone with a rare disease, and how these can differ from those of other caring populations who often receive more support.

- The narrative synthesis or rather the methodology part does not clearly indicate how the final themes were obtained. I suggest making the process of synthesis more explicit by including another table where this becomes clear to the reader.

The table entitled "Supplementary file 1 – Data extraction table" includes a column labelled, 'identified caregiver needs'. The needs extracted from the included papers and presented in this column were grouped together under headings which then became the themes. A sentence has been added to p5 lines 133-136 which hopefully make this process clearer.

The identified caregiver needs were extracted from the included papers [JM] and presented the data extraction table [AMcK, JM & LL] (Supplementary file 1). From this, common needs were grouped

together under headings which became the themes for this review. The themes and content for the themes was agreed between the research team [AMcK, JM & LL].

- Finally, you made some interesting discoveries that could be informative for policymakers and practitioners, but do not further elaborate on them nor fully include them in the implications/or strengths of your papers. I would advise you to make your contribution more explicit by adding these into the strength sections. You made some interesting suggestions regarding the possibility of creating a manual both online and hardcopy and this could be stressed out as a valuable implication for practice. I am attaching a pdf of your manuscript with some comments and suggestions that you might want to consider. Best Regards

Thank you. We have made the following changes on P10 lines 302-305:

“The insights that are gained through future research in relation to the impact on caregivers’ social lives should be considered in the priorities and strategic directions for clinical services. Interventions to support them following a diagnosis would be useful, delivered either in hardcopy or online.”

Comments from pdf

1. In the original article, this estimate is based on the continent of Europe. Here you refer to the specified population without mentioning that the estimate refers to Europe. In a sense, the original article is misinterpreted

Thank you for this. I appreciate that this detail should have been included so the phrase ‘In Europe..’ has now been included in this sentence on p2, line 58:

“In Europe, rare diseases (RDs) are those which affect less than 1 in 2000 people in a specified ...”

2. In the original article [2] this estimate is 350 rather than 300. It’s a rather large number to neglect.

Thank you, we have updated this figure to reflect the world’s population in August 2022 with a population prevalence of ~6% having a rare disease. This figure has been amended on page 2, line 60.

“...affecting approximately 450 million people globally^{2 3}. RDs result in a wide variety of healthcare needs...”

3. could you clarify this by giving an example what multi-organ systems refer to?

Thank you for this comment, an example has now been given on P2, lines 61-62

“such as respiratory complications, the circulatory system, muscular system, digestive system, and ...”

4. I would advise to define the care recipient population here in terms of age. It is not clear if the focus is on caregivers of young population/children with RD or adults. If the focus is on overall population of individuals living with RD, it might be difficult to precisely define the needs without grouping them per age range.

Thank you for this comment. As we did not put an age restriction on our inclusion criteria for studies, we are looking at all ages within this cohort. Although there will, of course, be differences in terms of the needs of caregivers in age groups, we found there were many more similarities and common needs which is what we were focused on in this review.

5. remove , and add "and" instead

This has been amended on p3 line 68

disease and is influenced by behavioural and social factors

6. It seems that on several occasions terms families and caregivers are used interchangeably. However, family does not necessarily mean caregiver, and a caregiver

does not necessarily mean family. I would suggest a clear distinction between family members and informal caregivers.

Thank you for making us aware of this potential confusion in the main text. An explanation has now been added on P3 lines 67-69 to make it clearer about common caregiving arrangements for someone with a rare disease.

“Although not always the case, family often plays a pivotal role in a person’s adjustment to chronic disease and is influenced by behavioural and social factors ⁸. A family member frequently takes on the role as caregiver or a person with a close relationship as an informal caregiver.”

7. Interestingly 50% of the patients with RD are diagnosed within four months, making this statement relative. I would suggest to include as an addition to this statement a general statistics regarding the diagnosis time span. This is available in [13].

Thanks for raising this comment. The average time for a diagnosis of a rare disease in Northern Ireland is 5 years. The UK Rare Diseases Framework prioritises getting a faster diagnosis for patients. P3 lines 75-76

(the average time for a diagnosis of a rare disease in Northern Ireland is 5 years).

8. It would be useful to provide an example on how the "care and services needs" are "associated with poor quality of care".

Thank you, an example has been added on page 3 line 80-81

“...for example lower satisfaction with healthcare services and care coordination...”

9. I would suggest to mention early on in the paper that the focus is on informal caregivers of children. This is the first indicator that this work is focusing on informal caregivers of children living with RD

Thank you, hopefully the explanation added to p2 as above has helped to explain this better. The rapid review focused on the needs of all carers of persons with a rare disease, not just parents. While we appreciate that caring for people of different ages may reflect different challenges, our approach aimed to capture the range of needs of caregivers of persons with rare diseases in the published literature to inform the development of tools to support carers and to influence local policy decisions.

10. it is still not clear if the focus is on overall individuals with RD or children. The issue becomes even more complex when the "informal caregiver" term and "family" are used interchangeably.

Thank you, an explanation has been added to P4 lines 101-104 to explain that the focus is on 'people with a RD'. No age restrictions or focus within this group.

“This review focuses on informal caregivers for people living with a RD. We were particularly interested in their holistic needs when caring for this unique population and how these may differ from other caring populations where there is often more support available e.g. carers of people with dementia or cancer.”

‘Family’ has been replaced with ‘informal caregiver’ on p6 line 184.

Previous research has shown that informal caregivers do not have enough time for themselves, ³²

11. please define informal caregiver early on in the paper

Thank you, as above this has now been defined on p4 lines 101-104.

This review focuses on informal caregivers for people living with a RD. We were particularly interested in their holistic needs when caring for this unique population and how these may differ from other caring populations where there is often more support available e.g. carers of people with dementia or cancer.

12. was there age range set for the studies?

There was not an age range set for the studies as although there are slight differences between age groups, there will be many more similarities and common needs which is what we were focused on in this review.

13. I do not understand still if this inclusion criteria means that the review focused on informal caregivers of adult individuals living with RD, and if so, why paper reflects on parents of children with RD earlier?

We have now provided additional information to help minimise this confusion p4 lines 112-115.

The criteria for inclusion were articles that address caregiving for people living with an RD. Articles on caregiving alone or RD alone were excluded. Primary research studies and systematic review were considered for inclusion as shown in Table 1.

14. The process of obtaining the overall themes is not very clear, nor what led to the formation of concrete themes. A sample table demonstrating part of the process could be helpful in understanding how were the final themes obtained

The table entitled 'Supplementary file 1 – Data extraction table', includes a column labelled, 'identified caregiver needs'. The needs extracted from the included papers and presented in this column were grouped together under headings which became the themes. A sentence has been added to p5 lines 133-136 which hopefully make this process clearer.

"The identified caregiver needs were extracted from the included papers [JM] and presented the data extraction table [AMcK, JM & LL] (Supplementary file 1). From this, common needs were grouped together under headings which became the themes for this review. The themes and content for the themes was agreed between the research team [AmcK, JM & LL]."

15. I would advise to define the type of support you deem beneficial, in line with the literature on the existing support for informal caregivers including support groups, respite care, employment arrangements, etc.

Examples have now been added to p6, line 188-189

"...for example from support groups, respite care and employment arrangements."

16. It would be useful to provide the examples of adverse impacts you refer to (e.g., health, mental health, financial, social, etc)

This is a good idea, thanks, examples have been added to p7 line 195

"...for example, financial and mental health ..."

17. This is an excellent point that can also be included in the implications

Thank you, this has now been included on p10 lines 302-305

"Interventions to support them following a diagnosis would be useful, delivered either in hardcopy or online ..."

18. Considering that caregivers and family members have been used interchangeably, relying on family members would inevitably mean simply relying on other caregivers? In that sense, are these other caregivers also informal caregivers or do they act as secondary caregivers?

Thank you, I can see how this sentence is confusing and it has now been reworded on p7, lines 206-207

"All too often, caregivers have little time to themselves and lack appropriate support and time for respite ..."

This is an excellent point, and should be further elaborated in the implications section

Thank you, this has been added to p10 lines 302-305

“The insights that are gained through future research in relation to the impact on caregivers’ social lives should be considered in the priorities and strategic directions for clinical services.”

19. Does this refer to the family relationship between the caregiver-care recipient or to the relationships among family members of the care recipient?

Thank you for making us aware that this sentence was not clear but has been amended on P10 lines 302-305

“The insights that are gained through future research in relation to the impact on caregivers’ social lives should be considered in the priorities and strategic directions for clinical services.”

20. There are numerous existing interventions for supporting informal caregivers, maybe you could include this as well and add that potentially caregivers lack knowledge or awareness of the existing support tools. Arguably, promotion of the existing validated tools could also be beneficial as the development of new approaches.

This is a good point to consider and this information has been added to p8 line 234-236

“Depressive symptoms are often associated with caregiving burden and therefore there is a need to develop interventions in addition to promoting the existing validated tools for caregivers considering their special needs.”

21. I wonder if this is psychological or social need?

We appreciate that ‘peer support’ could be both a psychological or a social need but as we are referring to emotional support in this instance, we have left in this section of the manuscript.

22. I think this is both information and communication need

Thank you, this is a good point and we appreciate that adding communication is more meaningful.

This has been amended on

p1 line 27, needs, financial needs, psychological needs, information and communication needs and

- P6 line 170 - Information and communication needs

and p8 line 245 - *Information and communication needs*

23. In my opinion this can also fall under the communication need. To clarify this, it would be useful, as already suggested, to include a sample of your work in obtaining the final themes. This would help reader understand how the final themes were distinguished

Please see above response to question 14.

24. I think you also have some material you could add in the implications section or further elaborate strength part in line with the previous comments referring to implications for practice and policymakers

Thank you for this suggestion, hopefully this has been addressed when responding to the points above from reviewer 3’s more general comments on the manuscript.

VERSION 2 – REVIEW

REVIEWER	Petrovic, Milica Catholic University of the Sacred Heart, Psychology
REVIEW RETURNED	25-Sep-2022
GENERAL COMMENTS	Thank you for responding to my previous suggestions. I think this version of the manuscript is a much-improved version but I still advise a few minor edits. Please see attached file.

	I think the informal caregiver is still not clearly defined and a distinction has not been made between the family members and the informal caregiver. In that sense, informal caregivers and parents are still being used interchangeably. Within a household, both parents are not necessarily informal caregivers. In fact, one parent could be an informal caregiver (e.g., shifting to part-time work, taking on all caregiving duties, changing his/her usual life routine) while another can maintain the role of the parent (e.g., relying on the spouse to provide primary care and neglecting additional caregiving duties). It is important to define informal caregivers clearly and early on in the paper without confusing or using the terms parents and informal caregivers interchangeably since the needs and experiences regarding RD of a parent who hasn't assumed caregiving might be much different than those of the parent who has assumed the role of the informal caregiver. I made several suggestions in the pdf file which require minor changes. Well done and great improvement.
--	---

VERSION 2 – AUTHOR RESPONSE

Reviewer: 3

Dr. Milica Petrovic, Catholic University of the Sacred Heart

Comments to the Author:

Thank you for responding to my previous suggestions. I think this version of the manuscript is a much-improved version but I still advise a few minor edits. Please see attached file.

I think the informal caregiver is still not clearly defined and a distinction has not been made between the family members and the informal caregiver. In that sense, informal caregivers and parents are still being used interchangeably. Within a household, both parents are not necessarily informal caregivers. In fact, one parent could be an informal caregiver (e.g., shifting to part-time work, taking on all caregiving duties, changing his/her usual life routine) while another can maintain the role of the parent (e.g., relying on the spouse to provide primary care and neglecting additional caregiving duties). It is important to define informal caregivers clearly and early on in the paper without confusing or using the terms parents and informal caregivers interchangeably since the needs and experiences regarding RD of a parent who hasn't assumed caregiving might be much different than those of the parent who has assumed the role of the informal caregiver. I made several suggestions in the pdf file which require minor changes. Well done and great improvement.

Many thanks for your comments, we are glad you feel this version of the manuscript is much improved. Please see below where each of your comments have been addressed individually.

*****Please also see attached comments*****

I would advise to merge these two sentences into one.

Thank you for making this suggestion. These 2 sentence have been merged to one on line 61-63.

RDs result in a wide variety of healthcare needs stemming from the involvement of multiple organ systems, such as respiratory complications, the circulatory system, muscular system, digestive system, and central nervous system.

I would suggest rewriting this sentence. Caregivers are defined as individuals who provide care for a person in need/care recipient. On the other hand informal caregivers are defined as family members or close others who provide care for the person in need with no financial benefit in return. In essence, formal caregivers/ or caregivers are a category of professionals or semi-professionals who provide care with financial benefit in return while informal caregivers are close others, family members or even friends who provide care but have no such benefits in return (in most European countries).

Many thanks, this sentence has been replaced with your suggestion (lines 69-75) which we appreciate is a much more comprehensive explanation of an informal caregiver.

Caregivers are defined as individuals who provide care for a person in need/care recipient. On the other hand informal caregivers are defined as family members or close others who provide care for the person in need with no financial benefit in return. In essence, "formal caregivers" or "caregivers" is used to denote a category of professionals or semi-professionals who provide care with financial benefit. In contrast, "informal caregivers" are typically family members or friends who provide care but have no such benefits in return (in most European countries).

I would suggest to split this sentence in two, it is rather confusing.

Thank you for this suggestion, this sentence has now been split on lines 83-86.

Research on the experience of having a RD indicates that care and service needs are often not based on the severity of the health condition. Rather, they are associated with poor quality of care and barriers to access leading to less satisfaction with healthcare services and care coordination^{15 17}.

It might be my error, but I could not find this table. Table 1 that I can see does not show concretely the number of primary research studies and systematic reviews but rather a simplified list of databases and number of papers retrieved. This is perfectly fine, but in that case consider correcting this statement.

Thank you, this was an error on our part. The statement has been corrected on lines 119-120.

A simplified list of databases and the number of papers retrieved is shown in Table 1.

You might also consider editing this in the Figure 2 where it is included as "information need"

Thank you for bringing this to our attention. Figure 2 has been edited to include 'information and communication'. The new figure has been uploaded for submission.

Caregivers and informal caregivers are two distinct categories although here have been used interchangeably, as well as "family caregiver". I think it is very important to remain consistent when using "informal caregivers" especially since the aim of this paper is to explore the needs of informal caregivers concretely and not caregivers overall.

Thank you, this has now been edited on line 189.

Informal caregivers are often forced to miss work or school days due to the demands of their role,

I would argue that caregivers who are professional or semi-professional do not experience social isolation. Therefore, it is important to use the clear distinction consistently with "informal caregivers"

Thank you for raising this point. 'Informal' has been added to line 209 as below.

Social isolation is common among informal caregivers with the impact on personal relationships being a reoccurring theme ^{4 5}. All too often, informal caregivers have little time to themselves and lack

please check this sentence once again

Thank you for making us aware of this, this sentence has been rewritten as below on line 310-312.

It is important that awareness is raised about the range of support options that are available from health and social care providers, charities and/or support groups for informal caregivers of people with an RD ²⁷.

VERSION 3 – REVIEW

REVIEWER	Petrovic, Milica Catholic University of the Sacred Heart, Psychology
REVIEW RETURNED	19-Oct-2022
GENERAL COMMENTS	Thank you for addressing my suggestions. The paper is well structured, keywords are clearly defined, methods are well explained, findings are appropriately presented, and conclusions made are relevant to the existing literature and current findings.

VERSION 3 – AUTHOR RESPONSE

Reviewer report

- Please include, as a supplementary file, the precise, full search strategy (or strategies) for all databases, registers and websites, including any filters and limits used.

Thank you for this comment. The search strategy has now been included as a separate file – Supplementary file 1.

- We have noticed you have uploaded two Supplementary File 1 in ScholarOne as the heading inside file is also 'Supplementary File 1'. Please upload only one Supplementary File 1 in ScholarOne. If there's Supplementary File 2, please cite this in the main document and put heading inside your file 'Supplementary File 2'.

Thank you, apologies for this oversight. The heading has been changed within Supplementary file 2. We have highlighted in the text where reference is made to each supplementary file. P4 line 114, p5 line 137, p5 line 160

combined terms 'informal caregiver*' and 'rare disease*' (Supplementary file 1). All searches were extraction table [AMcK, JM & LL] (Supplementary file 2). From this, common needs were grouped

each source summarised from the completed data extraction table (Supplementary file 2).

- We have noticed that your Ethics Approval Statement stated "not applicable" which is not allowed. Please provide the reason why Ethics Approval is not necessary. (e.g. This study does not involve human participants and ethical approval was not required.) Please also ensure that the Ethics Approval Statement in the main document and in ScholarOne should be the same.

Thank you, this statement has been updated on the main manuscript and in ScholarOne. P10 lines 318-319.

This study does not involve human participants and ethical approval was not required